# Circadian Clock Regulation of Hepatic Energy Metabolism Regulatory Circuits

**DOI:** 10.3390/biology8040079

**Published:** 2019-10-19

**Authors:** Ann Louise Hunter, David W. Ray

**Affiliations:** 1Centre for Biological Timing, Department of Diabetes, Endocrinology and Gastroenterology, Faculty of Biology, Medicine and Health, University of Manchester, Manchester M13 9PT, UK; louise.hunter@manchester.ac.uk; 2NIHR Oxford Biomedical Research Centre, John Radcliffe Hospital, Oxford, UK and Oxford Centre for Diabetes, Endocrinology and Metabolism, University of Oxford, Oxford OX3 9DU, UK

**Keywords:** circadian clock, biological rhythms, energy metabolism, liver, REV-ERBα, cryptochrome

## Abstract

The liver is a critical organ of energy metabolism. At least 10% of the liver transcriptome demonstrates rhythmic expression, implying that the circadian clock regulates large programmes of hepatic genes. Here, we review the mechanisms by which this rhythmic regulation is conferred, with a particular focus on the transcription factors whose actions combine to impart liver- and time-specificity to metabolic gene expression.

## 1. Introduction

The circadian clock maintains homeostasis in the face of rhythmic environmental changes and as such, regulates critical physiological processes, including inflammation and energy metabolism. Rhythms of hepatic metabolism have been especially well-studied, as interest grows in the potential of circadian medicine to have therapeutic applications for disorders of energy metabolism and storage. Here, we summarise the current understanding of the mechanisms by which programmes of metabolic genes are under circadian regulation in the liver, particularly by core clock transcription factors.

## 2. Rhythmic Control of Metabolic Gene Expression

### 2.1. The Circadian Clock

In mammals, there is a network of cellular clocks that are coordinated, or synchronized, by the central clock in the suprachiasmatic nucleus of the hypothalamus. The suprachiasmatic nucleus of the hypothalamus functions as the dominant pacemaker tissue, responsive to light via the retinohypothalamic tract. This central clock entrains clocks in peripheral tissues by autonomic and endocrine signals. Entrainment refers to the synchronisation of a rhythm to a fixed time signal, in the absence of entrainment, rhythms are free-running, and as the endogenous circadian clock period is not exactly 24 h, this results in separation of the endogenous circadian phase from that of the external environment through time. Peripheral tissue clocks are sensitive to phase setting by other stimuli (zeitgebers) and are highly responsive to feeding time [1,2,3] and glucocorticoids [4], such that altered feeding times or dexamethasone administration can reset peripheral clocks. Synchrony between central and peripheral clocks promotes harmony between the behaviour of the organism (activity, rest, feeding patterns) and for example, its immune response and nutrient utilisation. A further layer of desynchrony is conferred by shift work in which the internal clock phase is non-aligned with the external clock phase, resulting in attempting to sleep at a non-permissive circadian phase, or eating against the clock resulting in indigestion [5].

At a molecular level, the clock comprises a transcription-translation feedback loop (TTFL) with a periodicity of approximately 24 h (Figure 1). The circadian locomotor output cycles kaput (CLOCK) and brain muscle Arnt-like protein 1, Arntl (BMAL1) proteins are the forward or activating limb of the clock and initiate transcription by binding as a heterodimer to E-boxes (CACGTG) in the promoters of multiple clock-controlled genes [6]. These core-clock genes include the *Period1/2/3* (PER) and *Cryptochrome1/2* (CRY) genes, whose protein products serve to negatively regulate CLOCK and BMAL1 and their transcriptional targets [7,8]. *Per1* and *Per2* expression are further modulated by the basic helix-loop-helix factors DEC1 and DEC2, and by the basic leucine zipper transcription factor E4BP4 and its counterbalance transcription factor DBP [9]. Another critical regulatory limb of the clock comprises REV-ERBα/β (NR1D1/2) and RORα/β/γ (retinoic acid receptor-related orphan receptor, NR1F1/2/3). BMAL1 and CLOCK are also repressed by the recently-described CHRONO protein [10]. The TTFL generates and maintains cell-autonomous rhythms of transcription. Across all tissues studied, more than 80% of the mammalian transcriptome shows rhythmic expression [11], but rhythms show marked tissue-specificity [11,12], thus a transcript which cycles in one tissue may be arrhythmic in another. This rhythmic regulation is highly advantageous in organs such as the liver, where metabolic processes can be matched to the activity and feeding behaviour of the organism, and potentially incompatible cellular metabolic processes (for example, a cellular reductive vs oxidative environment) can be temporally segregated.

### 2.2. Layers of Gene Regulation

This tissue-specificity demonstrates that the core clockwork is not the only determinant of rhythmic gene expression. Gene expression is regulated by transcription factor activity at promoter regions, activity at distal regulatory regions (enhancers) and chromatin architecture [13,14]. Chromatin architecture encompasses chromosome accessibility, modifications to the histone proteins (which together with DNA comprise chromatin), and the three-dimensional organisation. Chromatin is typically arranged into nucleosomes (the repeating unit of DNA wrapped around histones akin to a string of beads). Active transcription is associated with open, accessible chromatin (euchromatin), whilst inactive areas of the genome are associated with tightly packed, inaccessible chromatin (heterochromatin). Certain transcription factors will only bind the genome at sites of open chromatin, whilst others have been shown to bind nucleosomes and trigger chromatin opening (pioneer factors). The histone proteins contain lysine residues which can be modified; different modifications are associated with different transcriptional states. H3K27ac (acetylation of lysine 27 on histone H3) is associated with active enhancer elements; H3K27me3 (trimethylation of lysine 27 on histone H3) is associated with the formation of heterochromatin and gene inactivation. Furthermore, transcriptional regulation does not take place in two dimensions, forwards and backwards along a length of DNA. Rather, the three-dimensional organisation of the genome permits sites some distance from each other to interact, being brought close together in loops of chromatin. Current understanding [15] suggests that loop formation is at the smaller scale end of organisational structures, extending upwards to chromosome compartments, and even positioning of individual chromosomes. Enhancer-promoter loop interactions occur more frequently within topologically-associating domains (TADs), which are bounded by insulator elements.

Any of these factors above can confer rhythmic regulation to gene expression. In fact, only 22% of oscillating genes are thought to be rhythmic due to de novo transcription [16], the remainder of the regulation resulting from post-transcriptional modification. A circadian rhythm of chromatin accessibility has been observed in mouse liver [17]. There are rhythms to histone modification [16] (again demonstrated in mouse liver) paralleling rhythmic activity of histone acetyltransferases and methyltransferases [18,19,20]. Time-of-day differences in loop formation have also recently been demonstrated [21,22]. 

In terms of conferring tissue-specific rhythmicity, current thinking suggests that patterns of chromatin accessibility, determined by tissue-specific pioneer factors, produce cistromes (the genome-wide binding profiles) of key clock factors distinct to different tissues. These can show surprisingly little overlap. A recent study has found only 6% of BMAL1 binding sites to be common to liver, kidney and heart [22], whilst only 183 REV-ERBα binding sites have been reported to be common to liver, white adipose tissue and brain [23]. One would predict that circadian loop formation is also tissue-specific, but this work has yet to emerge.

### 2.3. Circadian Control of Hepatic Gene Expression

Approximately 10% of the mouse liver transcriptome is reported to show rhythmic expression (the most recently published liver dataset detected 2029 genes with circadian oscillation [24]). In peripheral tissues such as the liver, disruption of the core molecular clock does not necessarily result in arrhythmicity, with evidence of newly rhythmic genes emerging in the absence of *Bmal1* [25], but loss of BMAL1 in the liver reduced the amplitude of the oscillations, and was accompanied by changes in DNAse1 footprint patterns marking loss of BMAL1:CLOCK heterodimer action [17]. Furthermore, restoration of the liver clockwork in an otherwise arrhythmic animal (*Bmal1*^−/−^ with liver-specific *Bmal1* reconstitution) only partly restores the rhythms seen in wild-type animals [24]. These findings underscore the importance of systemic cues in acting as zeitgebers to the liver clock, with insulin [3], the gut microbiome [26] and distal inflammation [27], all having the demonstrated ability to do so. Whilst glucocorticoids (administered in supraphysiological doses) have been shown to synchronise peripheral clocks [4], the role of endogenous glucocorticoids in regulating liver rhythmicity is unclear. In fact, although there is evidence that these signals can serve as zeitgebers to peripheral clocks, the evidence that they are important in physiology is rather weak.

In the livers of intact animals with normal behaviour, there is a coordinated rhythm of CLOCK:BMAL1 recruitment to the genome during the day, accompanied by histone acetylation and RNA polymerase II recruitment, which triggers a peak phase of transcription in the early part of the night [14,24]. As the translation of the repressive CRY and PER proteins increases, CLOCK and BMAL1 levels fall and transcription is repressed [14]. REV-ERBα, also regulated by CLOCK:BMAL1, shows peak recruitment to the genome at ZT8–10 (late day, where ZT0 = “lights on”) in liver [28,29], with a nadir at ZT20–22 (late night). This constitutes another negative feedback loop.

### 2.4. Hepatocyte Nuclear Factors

Following the theory that tissue-specific chromatin accessibility dictates clock protein binding [22], one could hypothesise that liver-specificity to rhythms of gene expression is conferred by liver-specific transcription factors such as FOXA2 and the hepatocyte nuclear factors (HNFs). This question has yet to be directly tested experimentally, however. In both mouse and human adult liver, the hepatocyte nuclear factor (HNF) transcription factors regulate large programmes of transcription. Chromatin immunoprecipitation experiments have demonstrated broad associations of HNF4α (an orphan nuclear receptor) and HNF6 (a transcriptional activator and part of the ONECUT family) with transcriptionally-active genes in adult human hepatocytes [30]. In mouse liver, chromatin accessibility and ChIP studies have shown that liver-specific enhancers are maintained by FOXA2, which subsequently permits HNF4α recruitment to these enhancers [31]. Together with a repressive complex comprising HDAC3 and PROX1, HNF4α then exerts control over liver lipid homeostasis [32]. The cistrome of HNF6 is similarly associated with a large number of genes in mouse liver, with genes related to cytochrome P450 and steroid metabolism being found close to HNF6 binding sites, although it should be noted, that the HNF6 cistrome shows considerable sexual dimorphism [33]. HNF6 has also been proposed to have repressive control of lipid metabolism, with liver *Hnf6* deletion leading to lipid accumulation [34] through tethering and recruitment of REV-ERBα.

Work demonstrating the synchronising effect of glucocorticoids in the liver has found the HNF4α motif to be associated with synchronised genes and that the synchronising effect was partly lost in liver *Hnf4α* knockout [35]. This supports the idea that HNF4α plays a role in conferring liver-specific gene rhythmicity. A more direct role for HNF4α in regulating the clock has also been proposed: namely, that HNF4α may itself repress CLOCK and BMAL1. This was concluded from the demonstration of reduced *Clock* and *Bmal1* expression with *Hnf4α* overexpression and co-occurrence of HNF4α binding at CLOCK:BMAL1 sites [36], but this work has yet to be expanded.

## 3. Core Clock Repressors Which Regulate Hepatic Energy Metabolism

Over the course of the day-night cycle, the nocturnal mouse undergoes transitions between the inactive and active states and the fasted and fed states. As the key organ of metabolism, the liver can both produce glucose as an immediate energy supply (gluconeogenesis) and produce lipid as a longer-term energy store (de novo lipogenesis) (Figure 2A). It is perhaps not surprising that these processes are under circadian control, to confer temporal regulation in line with behaviour at the level of the organism (Figure 2B). Two of the core clock proteins which have repressive activity, CRY and REV-ERBα, are of particular importance in regulating these pathways. Both CRY and REV-ERBs can affect metabolic function by actions on the core circadian clock, but both can also act outside of the transcription-translation-feedback loop by acting independently as transcriptional repressors, such as the transrepression of glucocorticoid receptor protein (GR) by CRYs [37].

### 3.1. Cryptochrome Regulation of Hepatic Gluconeogenesis

The CRY proteins in mammals are related to blue-light receptors and photolyases found in plants. They play a key role in both the early and late repressive phases of the circadian cycle, with CRY1 repressing the CLOCK:BMAL1 heterodimer by binding directly to the PAS (Per-Arnt-Sim) domain [38]. Degradation of PER and CRY proteins is one of the factors determining circadian periodicity, and it is of note that phosphorylation of CRY by the energy sensor AMPK (AMP-activated protein kinase) [39] reduces its stability, thus creating a means by which nutrient availability can influence the clock. Decreased CRY stability under conditions of energy deprivation, causes increased amplitude and therefore, more robust circadian oscillations.

In mouse liver, activity at the fasting-responsive cyclic adenosine monophosphate (cAMP) response element (CRE), which regulates gluconeogenesis, is lowest at the night-day transition when CRY activity is highest [40]. This has prompted investigation of the role of CRY in regulating gluconeogenesis, and the finding that CRY prevents the accumulation of cAMP in response to glucagon stimulation, likely via the inhibition of the Gs alpha subunit, thus reducing gluconeogenic gene expression [40]. The authors propose that pharmacological intervention to stabilise CRY might be a means of reducing gluconeogenesis and indeed, the small molecule activator of CRY, KL001, has been shown to reduce gluconeogenesis in response to glucagon stimulation of hepatocytes in vitro [41]. CRY has also been found to promote degradation of the FOXO1 (forkhead box O1) transcription factor, which is a positive regulator of gluconeogenic genes such as *Pepck* and *G6pc*, thus suppressing gluconeogenesis further [42].

Separately, CRY has been found to associate with the glucocorticoid receptor C-terminus [37]. It appears by mechanisms unclear, that this allows CRY to repress GR-mediated transcription, with an increased transcriptional response to dexamethasone seen in CRY-deficient mouse embryonic fibroblasts, with a particular effect seen on transactivated target genes. Furthermore, glucocorticoid-activation of the *Pepck* gene shows time-of-day dependency being highest when CRY levels are lowest. Chromatin immunoprecipitation experiments do suggest that CRY can associate with the *Pepck* GRE (Glucocorticoid Response Element), in a dexamethasone-dependent manner. Chronic dexamethasone treatment in CRY double-knockout mice significantly impairs glucose tolerance, this effect is not seen in wild-type mice [37]. Taken together, this strongly suggests that CRY directly inhibits glucocorticoid-induction of gluconeogenesis.

### 3.2. REV-ERBα Regulation of Hepatic Lipid Metabolism

#### 3.2.1. REV-ERBα Is a Nuclear Receptor

REV-ERBα is a member of the nuclear receptor superfamily. Its endogenous ligand is haem, a cofactor important for enzymes involved in mitochondrial respiration [43]. Haem synthesis is rate-limited by the ALAS-1 enzyme, which is under control of the master metabolic coactivator PGC-1α (PPARγ co-activator 1α). PGC-1α can itself co-activate ROR to drive *Bmal1* and *Rev-erbα* transcription [44]. When haem binds to REV-ERBα, it induces the recruitment of the repressive NCoR/HDAC3 (nuclear co-repressor/histone deacetylase 3) complex, which represses PGC-1α, thus allowing haem to regulate its own production and prevent haem toxicity [45]. This is also one of the many links between REV-ERBα and energy metabolism, with PGC-1α being a key activator of gluconeogenesis [46]. 

REV-ERBα has properties which distinguish it from other nuclear receptors. It has not, for example, been found to form functional heterodimers with other NR1 family members at the RORE site, unlike RXR and PPAR [47]. Co-regulator proteins, which include co-activators such as SRC-1, SRC-2, SRC-3 and the CBP (CREB binding protein)/p300 family, share an LXXLL motif (where L is leucine and X is any amino acid), which mediates interactions with ligand-bound nuclear receptors through the C-terminal H12 helix of the nuclear receptor [48]. REV-ERBα lacks this H12 helix [49] and so cannot interact with co-activators, underlining its function as a constitutive repressor. Interestingly, CRY and PER proteins possess the LXXLL motif [6] and have been shown to form heterodimers with diverse nuclear receptors [37].

#### 3.2.2. Constitutive Repressive Function of REV-ERBα

The P-box in the DBD of NR1 receptors recognises the AGGTCA motif, typically found as repeats (direct/inverted/everted) with variable numbers of spacing nucleotides between. REV-ERBA binds WAWNTAGGTCA as a monomer (where WAWNT is an A/T-rich sequence immediately 5’ to AGGTCA) demonstrating activation of transcription in vitro [47]. Later experiments (alongside REV-ERBβ) could not reproduce this activation as a monomer [50]. Both REV-ERBα and REV-ERBβ can, however, suppress transcriptional activation by ROR proteins at WAWNTAGGTCA sites (hereafter referred to as ROR elements, ROREs) [51] (Figure 3A). By contrast, as a homodimer, REV-ERBα binds the RevDR2 site (direct repeat of AGGTCA, with a 2 bp spacer) stably and with high affinity, demonstrating repressive function [52] (Figure 3B). It is this homodimerisation which permits recruitment of the co-repressor NCoR [53,54]. In addition, two REV-ERBα molecules binding in close proximity (in this case to two WAWNTAGGTCA motifs arranged as everted repeats 20 bp apart) are also able to recruit NCoR and repress transcription [55]. It is the presence of two RORE sites, 26 bp apart, in the *Bmal1* promoter which is proposed to be the mechanism by which REV-ERBα exerts repressive control over *Bmal1* transcription [56].

Two further mechanisms of REV-ERBα-mediated gene repression have been proposed. One of these suggests that the displacement of ROR at RORE sites is not through competition but through facilitated repression. This model requires the co-activator SRC-2 (GRIP-1) [57]. The SRC-2 cistrome displays diurnal variation, and SRC-2 functions as a co-activator for the CLOCK:BMAL1 heterodimer, such that SRC-2 ablation in the liver disrupts hepatic metabolism [58]. In the activation phase of the circadian cycle, SRC-2 recruits PBAF members of the SWI/SNF chromatin remodelling family, to allow the formation of a BMAL1/ROR complex, promoting maximal chromatin accessibility. REV-ERBA preferentially binds open chromatin, thus PBAF is gradually displaced over the course of the activation phase and gene expression falls. More recently, it has been shown that there are fewer functional intra-TAD interactions at the time of peak REV-ERBA action (ZT10) compared to the physiological nadir of REV-ERBA (ZT22) or in *Rev-erbα* knockout, thus it has been proposed that REV-ERBA opposes the formation of functional enhancer-promoter loops at its target genes [21].

#### 3.2.3. Dual Roles of REV-ERBα

REV-ERBα is proposed to have roles as a component of the core clock and as a master regulator of metabolism. Over recent years, evidence has suggested that these two roles are mediated by distinct mechanisms. Characterisation of the cistromes of REV-ERBα and REV-ERBβ in mouse liver has revealed overlap at sites associated with core clock genes, suggesting that both contribute to clock regulation [29] (similar overlap demonstrated separately in [59]). Motif analysis of REV-ERB binding sites from this dataset found the RevDR2 motif, but also motifs for other NRs (e.g., PPAR, CEBP, HNF4α). Comparison of the cistromes of REV-ERBα and RORα and of the transcriptomes of mice lacking either of these (*Rev-erbα*^−/−^ mice and *Rorα^Fl/Fl^Rorγ^Fl/Fl^* mice injected with AAV-Cre, respectively), suggests that the two proteins might compete for binding at RORE/RevDR2 sites associated with highly rhythmic genes [23].

This same study noted that many genes were *Rev-erbα*-dependent but did not associate with RORE/RevDR2 REV-ERBα binding sites. Rather, they associated with REV-ERBα binding sites containing the motif for the liver lineage-determining factor HNF6 (hepatocyte nuclear factor 6), which is thought to promote gene transcription by recruiting CBP/p300 to the genome. Using mice with a liver-targeted mutation in the REV-ERBα DBD, who concurrently had liver-targeted *Rev-erbβ* knockdown (AAV-Cre approach), it was hypothesised that REV-ERBα has both DBD-dependent and DBD-independent actions and that these are dissociable [23]. The DBD-dependent cistrome associates with RORE/RevDR2 motifs and with clock genes, whilst the DBD-independent cistrome associates with the HNF6 motif and lipid metabolism genes (Figure 3C). In the absence of HNF6, there is reduced REV-ERBα binding at sites associated with REV-ERBα-dependent genes [34]. As with liver depletion of HDAC3, liver depletion of HNF6 also leads to hepatosteatosis and up-regulation of the known REV-ERBα repressive targets SCD1, FASN, and ACC [34].

#### 3.2.4. REV-ERBα as a Metabolic Regulator

REV-ERBα is therefore proposed to mediate these DBD-independent metabolic effects by tethering to HNF6 and recruiting the NCOR/HDAC3 co-repressor complex [23,34]. The genome-wide binding sites of HDAC3, as determined by ChIP-sequencing, vary markedly by time-of-day [28]. At ZT10, more than 100-times more binding sites are seen for HDAC3 than at ZT22, and indeed, as this suggests, there are concurrent REV-ERBα and NCoR recruitment to these sites. Reduced acetylation of H3K9 (the target of the deacetylase activity of HDAC3) and decreased recruitment of RNA polymerase II, indicative of decreased transcription, are also seen. Interestingly, some of these binding sites are associated with genes for the enzymes of de novo lipogenesis, such as SCD1, FASN and ACC. In mice fed standard chow, depletion of HDAC3 in the liver leads to hepatosteatosis and increased de novo lipogenesis [28]. When the metabolic consequences of hepatic HDAC3 depletion are studied in greater detail, accumulation of small droplets of lipid is seen within hepatocytes of these mice [60]. This is a pathology distinct from the very large lipid droplets which accumulate around the central vein of the hepatic lobule of mice fed a high-fat diet, perhaps supporting further the idea of increased hepatic lipogenesis, rather than increased fat absorption or reduced peripheral uptake. The same investigators also report that liver HDAC3 depletion appears to reduce hepatic glucose output and suggest that metabolic intermediates are shunted towards lipid production, rather than glucose production, with the loss of HDAC3 (and by extrapolation, of REV-ERBα action).

The metabolic phenotype of mice lacking *Rev-erbα* is not as severe as that caused by depletion of HDAC3 or HNF6; this supports a role for *Rev-erbβ*, and indeed, in *Rev-erbα*^−/−^ mice who then have hepatic depletion of *Rev-erbβ*, the hepatosteatosis does become more severe and less NCoR/HDAC3 recruitment to the genome is seen [58]. Nonetheless, evidence of increased de novo lipogenesis, particularly towards the end of the rest period and during the active period (when the animal eats most), is seen in *Rev-erbα*^−/−^ mice [61]. Using indirect calorimetry, the authors also detect differences between knockout mice and wild-type controls in circadian patterns of fuel utilisation, with evidence of a preference for fatty acid use during the rest phase and glucose use during the active phase in the knockout mice [61]. They suggest too that this is due to imbalances in the shift of metabolic intermediates, with excess acetyl-CoA in the *Rev-erbα*^−/−^ mice driving excessive fat storage and ketogenesis.

Together, these studies build a picture of REV-ERBα/NCoR/HDAC3 repressive activity during the inactive period at sites regulated by HNF6 binding, which serves to downregulate de novo lipogenesis (potentially shunting the metabolic intermediates into gluconeogenesis). During the active feeding phase, this repressive activity falls, allowing lipogenesis and lipid accumulation. The absence of these repressive components therefore, produces a phenotype of increased de novo lipogenesis and hepatic lipid accumulation. Furthermore, an in vivo study with a REV-ERBα agonist has demonstrated the loss of fat mass with enhancement of REV-ERBα activity [62].

REV-ERBα is degraded by the proteasome having been ubiquitinated by the F-box protein FBXW7, this ubiquitination is itself triggered by phosphorylation by cyclin-dependent protein kinase 1 (CDK1) [63]. Modulating FBXW7 or CDK1 activity, therefore modulates rhythmicity of the core clock genes (FBXW7 ablation disrupts not only clock gene expression but also lipid metabolism [63], providing further evidence that REV-ERBα plays a role in regulating liver lipid metabolism). REV-ERBα degradation via SUMOylation and ubiquitination is also promoted by inflammatory processes [64], providing a means by which inflammation can modulate circadian amplitude.

## 4. The Role of Glucocorticoid Signalling in Circadian Control of Hepatic Energy Metabolism

Hepatic energy metabolism is also subject to regulation by rhythmic hormones, notably endogenous glucocorticoids (cortisol in humans, corticosterone in mice). Acute treatment with dexamethasone (a synthetic glucocorticoid) influences the expression of large programmes of metabolic genes in mouse liver [65], in keeping with the role of glucocorticoid hormones in the stress response in mobilising energy stores and increasing hepatic glucose output. Glucocorticoid production displays a diurnal rhythm, typically with a peak at the start of the active phase and a trough in the middle of the rest phase. Expression of the glucocorticoid receptor gene is rhythmic [66] and levels of the glucocorticoid receptor protein (GR) in hepatocyte nuclei, as quantified in proteomic studies [67], closely follow the rhythm of endogenous corticosterone. N-terminal phosphorylation marks associated with nuclear location of GR similarly peak at the start of the active phase, whilst phosphorylation marks associated with GR being cytoplasmic appear to be more abundant during the inactive phase [68].

Glucocorticoid action is context-specific [69,70,71] and indeed, a time-of-day effect on glucocorticoid action has been demonstrated [65], with many more genes in mouse liver being glucocorticoid-responsive during the day (inactive phase) than during the night. The mechanisms by which the circadian clock might confer time-specificity to GR action are being elucidated. The CRY proteins have been shown to negatively regulate glucocorticoid action [37] and it has been proposed that REV-ERBA facilitates circadian recruitment of GR to GC-responsive genes [65]. It has also been suggested that REV-ERBα can affect the nuclear localisation and the stability of GR [72]. 

This raises the attractive possibility that manipulating the timing of glucocorticoid administration could minimise the undesirable metabolic side effects associated with therapeutic glucocorticoid use. A study of chronic glucocorticoid administration has found that when given at ZT12, glucocorticoids up-regulate lipogenic genes to a lesser extent than at ZT0 [73]. Further work is needed in this area, however.

## 5. Summary

The liver plays a critical role in enabling the necessary switch between the fed and the fasted state. As such, gene expression programmes in the liver show a strong time of day dependence. The relative importance of the liver-autonomous circadian clock versus the systemic circadian system in generating these rhythms of gene expression and enzymatic function remains unclear. A number of key circadian transcription factors, including CRY and REVERBa, have been shown to play dominant roles in regulating either carbohydrate or lipid metabolic switches in the liver through time, and both these transcription factors have been targeted by small, drug-like molecules. A further manifestation of the circadian phenotype results from regulation of the hepatic nuclear receptor expressome, with actions identified both for lineage-determining nuclear receptors including HNF4A, and also signal-dependent receptors, such as the glucocorticoid receptor. Taken together, there is now abundant evidence for the importance of the hepatic circadian clock in driving energy metabolism, utilisation and storage and for the actions of systemic diseases, such as obesity, in affecting this process. It seems certain that new approaches to tackle the prevalent disease of over-nutrition in developed economies will require the liver clock to be harnessed for maximal efficacy.

## Figures and Tables

**Figure 1 biology-08-00079-f001:**
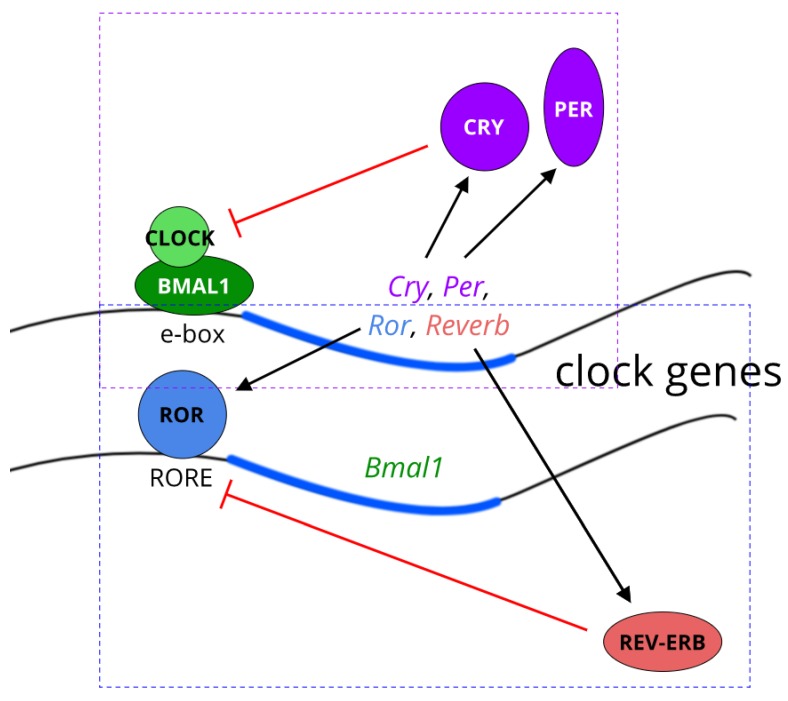
Cartoon of the core molecular clock. Simplified illustration of the core molecular clock comprising: the circadian locomotor output cycles kaput (CLOCK), brain muscle Arnt-like protein 1, Arntl (BMAL1) and the genes *Cryptochrome1/2* (CRY)/*Period1/2/3* (PER) transcription-translation feedback loop (TTFL) (purple dashed box; and the regulatory REV-ERB/ROR limb (blue dashed box).

**Figure 2 biology-08-00079-f002:**
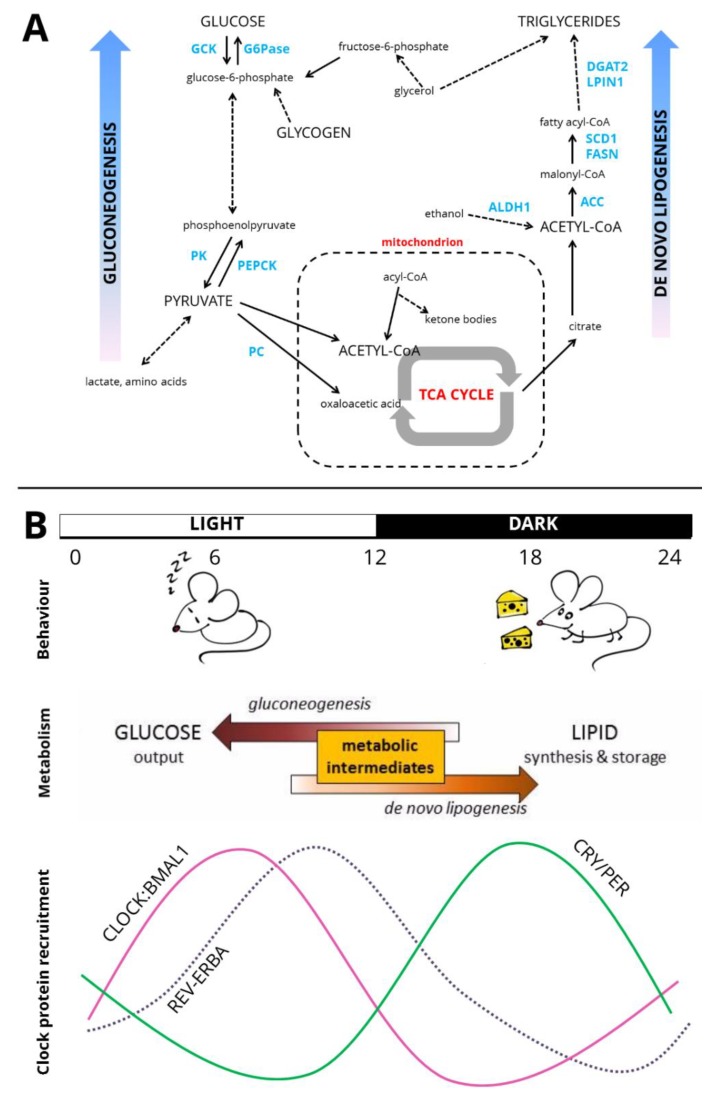
Circadian liver metabolism. (**A**) Pathways of hepatic carbohydrate and lipid metabolism. Key enzymes highlighted in blue. Truncated pathways are denoted by dashed arrows. GCK—glucokinase, G6Pase—glucose-6-phosphatase, PK—pyruvate kinase, PEPCK—phosphoenolpyruvate carboxykinase, PC—pyruvate carboxylase, TCA cycle—tricarboxylic acid cycle/citric acid cycle, ALDH1—aldehyde dehydrogenase 1, ACC—acetyl-CoA carboxylase, FASN—fatty acid synthase, SCD1—stearoyl-CoA desaturase 1, LPIN1—lipin 1/phosphatidate phosphatase, DGAT2—diacylglycerol O-acyltransferase 2. (**B**) Cartoon of circadian organisation across the organism. Over the light-dark cycle (shown here as 12:12), rhythms of behaviour are exhibited with mice inactive and relatively fasted during the day and active and feeding at night. Metabolic intermediates are shunted towards gluconeogenesis during fasting, and de novo lipogenesis when feeding. At the level of the genome, recruitment to the genome of core clock activators, the CLOCK:BMAL1 heterodimer, is greatest in the latter half of the day [14]. At night, the CRY and PER proteins, the repressive limb of the core clock, accumulate. REV-ERBA recruitment to the genome is greatest at ZT8–ZT10 [27,28].

**Figure 3 biology-08-00079-f003:**
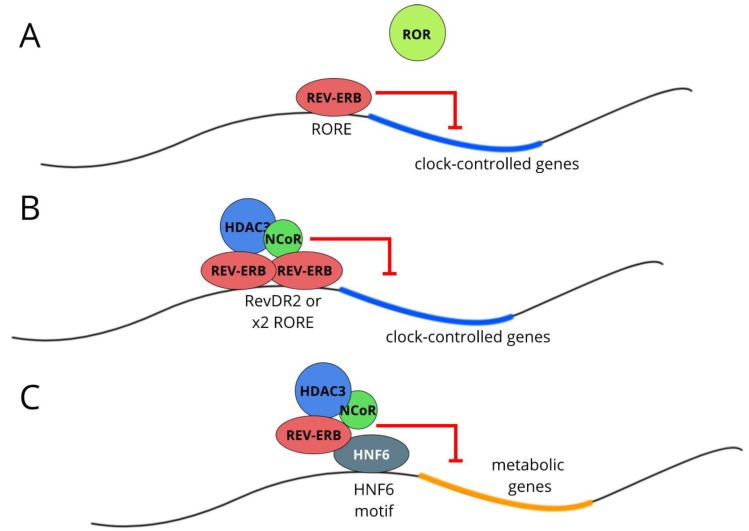
REV-ERB as a circadian and metabolic regulator. (**A**) It is proposed that REV-ERB displaces ROR at RORE sites and thus exerts rhythmic negative control over programmes of gene expression. (**B**) REV-ERB can also bind to the RevDR2 motif or two closely-spaced RORE motifs and, as a homodimer, recruit the NCoR/HDAC3 repressor complex. (**C**) REV-ERB repression of metabolic genes is proposed to be DBD-independent and require tethering of the REV-ERB protein to tissue-specific transcription factors, such as HNF6. REV-ERB then recruits the repressor complex as before.

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
