# Peer review of "Circadian Clock Regulation of Hepatic Energy Metabolism Regulatory Circuits"

_biology, 2019, doi:10.3390/biology8040079_

Round 1
Reviewer 1 Report
The manuscript by Hunter and Ray reviews the role of the circadian clock in the regulation of hepatic metabolism. In the first part it introduces the mammalian circadian clock with a particular emphasis on the various levels of transcriptional regulation and the largely unresolved question of how tissue specific oscillations are generated. The second part explains in detail the gene regulatory mechanisms that govern the circadian control of hepatic metabolism with a special focus on REVERBa. I quite enjoyed reading the manuscript and I think that most people with an interest in clock-dependent physiological regulation also would.
Some statements might benefit from a clarification:
Lanes 33-36: I believe that while each of the two sentences by itself is correct, the mentioned “dyssynchrony” induced e.g. by shift work is under most circumstances not a dyssynchrony between the central and peripheral clocks, as the previous sentence would imply, but rather a dyssynchrony between the circadian system as a whole and the behaviour of a person. For example, in a person working a fast forward rotating shift plan the central and peripheral clocks will not be uncoupled in such a short time span, they will not even phase shift a lot. Therefore, during night shifts eating and drinking happens at non-permissive times, which results in sleeplessness and digestion problems.
Lane 42 and Figure 1: The term “clock-controlled genes” is normally used to differentiate so called output genes that are regulated by clock transcription factors but which themselves are not part of the oscillator mechanism from clock genes, i.e. the genes for the clock transcription factors.
Lane 78-79: The paper by Koike et al uses the term “de novo transcription” to differentiate transcriptional from post-transcriptional regulation and not to differentiate gene regulation by transcription factors from regulation by chromatin and nuclear structure that the authors explain in the two preceding paragraphs.
Lanes 83-86: The context makes it sound like rhythmic regulation is mainly caused by loop formation and other chromatin mechanisms whereas it might be intended to be a more general statement about the advantages of rhythmic gene regulation. In the latter case I would move the sentence further up, e.g. lane 51.
Lanes 97-107: It might be worthwhile adding that for nearly all peripheral Zeitgebers only proof-of-principle studies exist and that it is not clear if all of them are functional in vivo, and even less, if some are more important than others.
Lanes 113-115: This is not a clear cut case but I would argue that REVERBa is regulated by the “normal” C/B-P/C limb but that it forms an”other main limb” together with the RORs.
Lane 220: A reference to Schmutz et al, Genes Dev 2010 could be included.
Lane 349-353: To my knowledge circadian hormones are normally substituted at those times of the day when they normally peak. Otherwise they don’t work well or even cause harm. As an outlook I would find it very interesting to learn more about the possibilities and limitations of such an approach for liver diseases. Do studies already exist, at least in animals?
Author Response
Lanes 33-36: I believe that while each of the two sentences by itself is correct, the mentioned “dyssynchrony” induced e.g. by shift work is under most circumstances not a dyssynchrony between the central and peripheral clocks, as the previous sentence would imply, but rather a dyssynchrony between the circadian system as a whole and the behaviour of a person. For example, in a person working a fast forward rotating shift plan the central and peripheral clocks will not be uncoupled in such a short time span, they will not even phase shift a lot. Therefore, during night shifts eating and drinking happens at non-permissive times, which results in sleeplessness and digestion problems.
We agree, and we have expanded and clarified this section.
Lane 42 and Figure 1: The term “clock-controlled genes” is normally used to differentiate so called output genes that are regulated by clock transcription factors but which themselves are not part of the oscillator mechanism from clock genes, i.e. the genes for the clock transcription factors.
We agree, and we have changed the text and the fig1 annotation. CRYs and PERs are clock genes.
Lane 78-79: The paper by Koike et al uses the term “de novo transcription” to differentiate transcriptional from post-transcriptional regulation and not to differentiate gene regulation by transcription factors from regulation by chromatin and nuclear structure that the authors explain in the two preceding paragraphs.
We have clarified this sentence to indicate that the missing circadian element is due to post transcriptional modification and regulation.
Lanes 83-86: The context makes it sound like rhythmic regulation is mainly caused by loop formation and other chromatin mechanisms whereas it might be intended to be a more general statement about the advantages of rhythmic gene regulation. In the latter case I would move the sentence further up, e.g. lane 51.
We have moved the sentence up as suggested.
Lanes 97-107: It might be worthwhile adding that for nearly all peripheral Zeitgebers only proof-of-principle studies exist and that it is not clear if all of them are functional in vivo, and even less, if some are more important than others.
We agree, and we have added a caveat to that effect.
Lanes 113-115: This is not a clear cut case but I would argue that REVERBa is regulated by the “normal” C/B-P/C limb but that it forms an”other main limb” together with the RORs.
We agree and we have changed the wording to indicate that the REVERB circuit is another negative feedback loop initiated by CLOCK:BMAL1.
Lane 220: A reference to Schmutz et al, Genes Dev 2010 could be included.
We were not sure that this addition was needed here. Did the referee intend to cite Guenther Schuetz?
Lane 349-353: To my knowledge circadian hormones are normally substituted at those times of the day when they normally peak. Otherwise they don’t work well or even cause harm. As an outlook I would find it very interesting to learn more about the possibilities and limitations of such an approach for liver diseases. Do studies already exist, at least in animals?
There are few studies, but the hormones work whenever given...eg GH by day, hydrocortisone at night, testosterone at night. However, there is a lack of a rigorous comparison on efficacy by time of day. We think that the lack of hard data makes it difficult to expand this section.
Reviewer 2 Report
The current review summarized molecular mechanisms of gene regulation by several circadian transcription factors. While the content is fine, the title is misleading. There isn't much discussion about how circadian clock impacts energy metabolism in the liver.
Author Response
The current review summarized molecular mechanisms of gene regulation by several circadian transcription factors. While the content is fine, the title is misleading. There isn't much discussion about how circadian clock impacts energy metabolism in the liver.
We can change the title to circadian clock control regulation of hepatic energy metabolism regulatory circuits. This recognises the bulk of the review relates to the control systems rather than energy metabolism per se.
Reviewer 3 Report
Hunter and Ray in their short review paper describe the role of the circadian clock in the regulation of genes involved in liver energy metabolism. The paper is timely and of interest to a broad audience. It is also reads quite well.
I have a few suggestions for improvement:
The paper very much focuses on transcriptional regulation (which is also mentioned in the abstract) and I think this could be better reflected in the title. Line 28: I would suggest inserting a sentence mentioning the network organization of cellular clocks in the very beginning to outline the content of the first two paragraphs. Line 39: Please explain free-run and entrainment. Otherwise, the notion about the ca. 24-h periodicity is confusing. Line 45: Please mention DBP as the counterbalance of NFIL3 (E4BP4) in the regulation of Per transcription. Also, please mention the official gene symbols and names at least once (Arntl for Bmal1, Nr1d1 for RevErba etc.). Figure 1: This figure could be improved by mentioning that the E-box belongs to Cry/Per/RevErb/Ror while the RORE belongs to Bmal1 (and then add a line from the Bmal1 gene to the BMAL1 protein). The the CCGs could be put to the right between both DNA lines as there are not only E-Box, but also RORE CCGs. Further, the inhibitory line from REV-ERB to CLOCK/BMAL1 is confusing as this inhibition is indirect (via regulation of Bmal1 transcription – s.a.). It would also be more complementary to the text if the two loops a clearly marked. Line 56-76: add some references here. Line 82 (and further places throughout the manuscript): after Ref 18 there is an "@" in my printout. Is this intentional? I do not find this for all references. Line 107: please mention this study: PMID16303750 Line 169+: it would be helpful to insert a short comment on clock vs. non-clock mediated effects of specific clock genes such as Crys and RevErb. Please also clearly distinguish between TTFL effects (i.e. via transcription) or non-TTFL effects (e.g. via direct interaction such as CRY/GR).
Author Response
The paper very much focuses on transcriptional regulation (which is also mentioned in the abstract) and I think this could be better reflected in the title.
We have changed the title.
Line 28: I would suggest inserting a sentence mentioning the network organization of cellular clocks in the very beginning to outline the content of the first two paragraphs.
we agree and have added a sentence.
Line 39: Please explain free-run and entrainment. Otherwise, the notion about the ca. 24-h periodicity is confusing.
We agree and we have added a sentence to this effect.
Line 45: Please mention DBP as the counterbalance of NFIL3 (E4BP4) in the regulation of Per transcription.
This is now added.
Also, please mention the official gene symbols and names at least once (Arntl for Bmal1, Nr1d1 for RevErba etc.).
This is now done.
Figure 1: This figure could be improved by mentioning that the E-box belongs to Cry/Per/RevErb/Ror while the RORE belongs to Bmal1 (and then add a line from the Bmal1 gene to the BMAL1 protein). The the CCGs could be put to the right between both DNA lines as there are not only E-Box, but also RORE CCGs. Further, the inhibitory line from REV-ERB to CLOCK/BMAL1 is confusing as this inhibition is indirect (via regulation of Bmal1 transcription – s.a.). It would also be more complementary to the text if the two loops a clearly marked.
We have revised the fig as suggested.
Line 56-76: add some references here.
We have done this.
Line 82 (and further places throughout the manuscript): after Ref 18 there is an "@" in my printout. Is this intentional? I do not find this for all references.
This is an error in formatting and is corrected.
Line 107: please mention this study: PMID16303750
this is a nice study and we happy to add the ref.
Line 169+: it would be helpful to insert a short comment on clock vs. non-clock mediated effects of specific clock genes such as Crys and RevErb. Please also clearly distinguish between TTFL effects (i.e. via transcription) or non-TTFL effects (e.g. via direct interaction such as CRY/GR).
We have added this sentence to the end of the opening para of section 2.